# What’s New in Cirrhotic Cardiomyopathy?—Review Article

**DOI:** 10.3390/jpm11121285

**Published:** 2021-12-03

**Authors:** Aleksandra Bodys-Pełka, Maciej Kusztal, Joanna Raszeja-Wyszomirska, Renata Główczyńska, Marcin Grabowski

**Affiliations:** 1Department of Cardiology, Medical University of Warsaw, 02-097 Warsaw, Poland; aleksandra.bodys-pelka@wum.edu.pl (A.B.-P.); maciekrm@gmail.com (M.K.); marcin.grabowski@wum.edu.pl (M.G.); 2Doctoral School, Medical University of Warsaw, 02-091 Warsaw, Poland; 3Liver and Internal Medicine Unit, Medical University of Warsaw, 02-097 Warsaw, Poland; joanna.wyszomirska@wum.edu.pl

**Keywords:** cirrhotic cardiomyopathy, liver dysfunction, cardiac dysfunction, portal hypertension, hyperdynamic circulation

## Abstract

Cirrhotic cardiomyopathy (CCM) is a relatively new medical term. The constant development of novel diagnostic and clinical tools continuously delivers new data and findings about this broad disorder. The purpose of this review is to summarize current facts about CCM, identify gaps of knowledge, and indicate the direction in which to prepare an updated definition of CCM. We performed a review of the literature using scientific data sources with an emphasis on the latest findings. CCM is a clinical manifestation of disorders in the circulatory system in the course of portal hypertension. It is characterized by impaired left ventricular systolic and diastolic dysfunction, and electrophysiological abnormalities, especially QT interval prolongation. However, signs and symptoms reported by patients are non-specific and include reduced exercise tolerance, fatigue, peripheral oedema, and ascites. The disease usually remains asymptomatic with almost normal heart function, unless patients are exposed to stress or exertion. Unfortunately, due to the subclinical course, CCM is rarely recognized. Orthotopic liver transplantation (OLTx) seems to improve circulatory function although there is no consensus about its positive effect, with reported cases of heart failure onset after transplantation. Researchers indicate a careful pre-, peri-, and post-transplant cardiac assessment as a crucial point in detecting CCM and improving patients’ prognosis. There is also an urgent need to update the CCM definition and establish a diagnostic algorithm for early diagnosis of CCM as well as a specific treatment of this condition.

## 1. Background

Cirrhotic cardiomyopathy (CCM) is a relatively new medical term. It was first described in 2005 during the World Congress of Gastroenterology in Montreal as characteristic changes in the circulatory system in patients with alcoholic liver cirrhosis. Initially, it was suspected that the toxic effect of alcohol was responsible for the pathogenesis of these changes. Subsequent studies have shown that the same haemodynamic abnormalities also occur in cirrhosis with a different aetiology [1,2].

Cirrhotic cardiomyopathy is a clinical manifestation of disorders in the circulatory system in the course of portal hypertension. It is characterized by impaired left ventricular systolic and diastolic dysfunction and electrophysiological abnormalities, especially QT interval prolongation. Unfortunately, due to the subclinical course, these irregularities are rarely recognized. The disease usually remains asymptomatic with almost normal heart function, unless patients are exposed to stress. Decompensation of the circulatory system may occur during liver transplantation, which is responsible for increased perioperative mortality due to cardiovascular causes [3,4].

In this review, we summarize the most important facts concerning cirrhotic cardiomyopathy and indicate the direction for further research.

## 2. Methods

We performed a review of the literature. The search was carried out using PubMed and ScienceDirect as data sources. We searched classical articles and clinical trials. The following keywords were used: “cirrhotic cardiomyopathy”. The inclusion criteria included articles that reported upon cirrhotic cardiomyopathy written in the years 1953–2021. Non-English articles were excluded.

### 2.1. Epidemiology and Diagnostic Criteria

The incidence of cirrhotic cardiomyopathy is not clearly defined. There was no correlation between the aetiology of liver failure and the incidence of cirrhotic cardiomyopathy. However, it has been proven that the prevalence of CCM depends on the severity of liver cirrhosis. Based on the Child–Pugh classification of hepatic insufficiency, the majority of CCM diagnoses are found in patients in Class C [5,6].

At the 2005 World Congress of Gastroenterology in Montreal, diagnostic and supportive criteria were proposed for cirrhotic cardiomyopathy. According to this definition, CCM symptoms include: systolic dysfunction (blunted increase in cardiac output on exercise, volume challenge or pharmacological stimuli or resting ejection fraction <55%), diastolic dysfunction (the ratio of early to late phases of ventricular filling or E/A ratio <1.0, prolonged deceleration time, prolonged isovolumetric relaxation time), supportive criteria: electrophysiological abnormalities, abnormal chronotropic response, electromechanical dyssynchrony, prolonged QTc interval, enlarged left atrium, increased myocardial mass, increased brain natriuretic peptide and pro-BNP, increased Tn I.

Since 2005, the progress in cardiac diagnostics greatly changed the concept of ventricular dysfunction and made the criteria out of date. In 2019, a multidisciplinary group consisting of cardiologists, hepatologist, and anaesthesiologists formed the Cirrhotic Cardiomyopathy Consortium and presented novel proposed criteria of CCM [7,8]. These novel diagnostic criteria approach CCM with contemporary cardiovascular imaging parameters with an emphasis on echocardiography. However, many additional criteria should be considered and assessed for their diagnostic use in the broad spectrum of CCM manifestations [7]. Table 1 contains a comparison of criteria from 2005 and 2019. Moreover, we added to the table our proposal for new parameters that should be included in the newly forming definition of CCM.

Due to the fact that the disease is usually asymptomatic, data on the prevalence of cirrhotic cardiomyopathy are very limited. Symptoms of hemodynamic cardiac dysfunction may occur when patients are exposed to stress. Some authors have shown that 50% of patients undergoing liver transplantation have developed some symptoms of heart dysfunction. However, about 7–21% of patients died of heart failure in the period after liver transplantation [9,10,11].

### 2.2. Pathophysiology

Pathomechanisms underlying cirrhotic cardiomyopathy are complex and not clearly understood. Moreover, their prevalence in the course of disease is also hard to determine. Cirrhotic cardiomyopathy is characterized by an impaired cardiac response to stress, which results from a combination of autonomic dysfunction, alterations in cell membrane composition, ion channel defects, and an overproduction of cardio depressant factors. A more detailed pathomechanism is presented in the following sections, and a diagram is presented in Figure 1.

#### 2.2.1. Cardiovascular Autonomic Dysfunction

One of the most common alterations in CCM is activation of the RAA system [12] and enhanced sympathetic tone [13], caused by progressive vasodilatation and a decrease in the effective arterial blood volume [14]. The main factor triggering autonomic overactivity seems to be a baroreceptor-reflex compromised by reduced blood pressure and volume [15,16].

An increased sympathetic tone accompanied by prolonged cardiomyocyte exposure to noradrenalin may result in myocardial injury, desensitization, and down-regulation of the ß-adrenalin receptor [17].

#### 2.2.2. Cell Membrane Modifications

Another important element in the pathophysiology of CCM is alterations in cell membrane structure. Decreased membrane fluidity and permeability along with an elevated level of membrane cholesterol may be caused by an increased serum concentration of cholesterol and a decreased concertation of lecithin-cholesterol acyltransferase (LCAT) [18]. These changes could interfere in the activation of many membrane-bound receptors and ion-channels, such as the ß-adrenalin receptor, calcium channel, Na^+^/K^+^ ATPase, Na^+^/Ca^2+^ exchanger, and Ca^2+^ ATPase [19,20]. Alterations in membrane ion transfers, especially Ca^2+^ and K^+^, may play an important role in inducing heart electrical abnormalities [21]. Moreover, sarcolemma fluidity modification could have an impact on the impaired stimulation of cardiac muscarinic receptors controlling sinoatrial node activity and atrioventricular conduction [22].

#### 2.2.3. Inflammation

Inflammation and its complications could trigger cardiac dysfunction, particularly in patients with decompensated cirrhosis. Several possible factors inducing inflammation are postulated, such as impaired intestinal permeability, delayed lymphocyte recruitment in mesenteric lymph nodes, and bacterial translocation leading to constant stimulation of the immune system [23,24]. In response to the circulating bacterial antigens, leukocytes produce and release cytokines and reactive oxygen species, which may impair myocardial function due to myocyte contractility dysfunction and extracellular matrix overgrowth [25].

#### 2.2.4. Humoral Cardiodepressant Factors

Endocannabinoids are bioactive signaling particles, increased serum concentration of which is observed in patients with cirrhosis, probably as a result of inflammation [26]. Their binding to the CB1 receptor reduced heart systolic function in animal models of CCM [27].

Another negative inotropic agent is nitric oxygen (NO) [28] and its production is also correlated with inflammation [29].

In the course of cirrhosis there is up-regulation in haem oxygenase, an enzyme producing carbon monoxide (CO), whose elevated serum concentration leads to blunted papillary muscle contractility [30].

Prolonged cardiomyocyte exposure to humoral agents such as CO, NO, endocannabinoids, and impaired intracellular calcium homeostasis may result in activation of cell apoptosis [22,31].

### 2.3. Clinical Features

CCM is mainly a subclinical syndrome. Signs and symptoms reported by patients are non-specific and include reduced exercise tolerance, fatigue, peripheral oedema, and ascites. Some of these features may not be easily distinguished from those of the underlying diseases [9].

Due to the splanchnic vasodilatation associated with cirrhosis, CCM is rarely manifested by symptomatic heart failure, although most patients with advanced liver disease are affected by the cardiomyopathy [32]. However, heart failure may be induced in the case of rapid normalization of peripheral resistances and blood pressure, as happens as TIPPS or OLTx are performed [33].

In the course of CCM there are several abnormalities associated with the systolic, diastolic, and electrical function of the myocardium.

Diastolic dysfunction is presumably the initial manifestation of CCM and precedes systolic dysfunction [34]. The mechanisms underlying its development remain unclear; however, some mechanisms, such as compromised intracellular calcium exchange, extracellular matrix abnormalities, increased myocardial stiffness, hyperaldosteronism, and cardiomyocyte hypertrophy, are postulated [22]. Doppler echocardiography is a method of choice for diastolic function assessment in patients with CCM. Abnormalities found in echocardiography suggesting diastolic heart dysfunction include [22,35]:-LA dilatation and increased LA volumes;-Increased LV diameters (but not volumes);-Increased thickness of the posterior wall of the LV and the interventricular septum;-Prolonged isovolumetric relaxation time (IVRT > 80 ms);-Prolonged deceleration time (DT > 200 ms);-E/A ratio < 1 (after age correction).

It is worth noting that current guidelines of the European Society of Cardiology (ESC) and the American Society of Echocardiography (ASE) propose more complex algorithms for diastolic dysfunction using a wider range of echocardiographic parameters than those mentioned above. This shows that the working definition of CCM has become much simpler and outdated [36].

Conventional pulse-wave doppler technique had limitations in patients with CCM (age, decreased preload and afterload, increased plasma volume) [37]. In addition, the mentioned criteria for CCM are mistaken when compared to current guidelines regarding diagnosis of diastolic dysfunction. For example, there are three ranges of the E/A ratio, which require additional diagnostic steps, mainly with tissue doppler imaging (TDI). This allows to calculate the early diastolic mitral annular velocity from the septal and lateral side (e’ septal, e’ lateral) as well as the average early diastolic mitral annular velocity (e’ av.). The American Society of Echocardiography has suggested that diastolic dysfunction is characterized by the presence at least three abnormal parameters, such as septal e’ velocity <7 cm/s or lateral e’ velocity <10 cm/s, an E/e’ average ratio >14, tricuspid regurgitation velocity >2.8 m/s, and LA volume indexed for body surface area (LAVI) ≥ 34 mL/m^2^ [22].

There are contradictory results in the case of a correlation between the diastolic dysfunction grade and liver failure stage. Some papers confirmed the significance of this correlation [38,39,40], whereas others did not find these association significant [41]. However, there is general agreement that the diastolic disfunction grade is not related to the etiology of liver disease, although there is a positive correlation with the Child–Pugh and MELD scores [38]. There have been other incoherent research findings regarding the correlation between diastolic disfunction grade and patients’ mortality [22,38].

Systolic dysfunction in cirrhotic patients is mostly latent, owing to low peripheral resistances and hyperdynamic circulation. Their cardiac pressures remain within normal limits at rest due to decreased afterload, as well as increased cardiac output and normal ejection fraction (EF) [22]. Although the left ventricular systolic function remains normal at rest, minor abnormalities could be unmasked under stress conditions, due to exertion, or by modern echocardiographic techniques at rest [37]. Patients with cirrhosis have documented blunted responsiveness to volume and postural changes, exercise or pharmacological agents (including dobutamine)—the increase in CO, EF, and HR is lesser than expected [11].

In the course of CCM dysfunction in the electrical activity of the myocardium may occur, manifesting as a prolonged QT interval, electromechanical uncoupling, and chronotropic incompetence.

QT interval prolongation is an integral part of the CCM definition. QT interval prolongation (calculated by the Bazett formula) is observed in 40–55% of patients with liver diseases [42]. Some studies have shown that the prolongation of the QT interval is independent of the etiology of liver disease, but is related to the severity of the disease and may have prognostic significance [43,44]. However, it is not associated with higher mortality rates in patients after OLTx. QT interval prolongation establishes pathomechanisms that include autonomic dysfunction, exposure to humoral agents from portal circulation by TIPPS, and abnormalities in cardiomyocyte calcium homeostasis [22]. Hyperkinetic circulation and the high prevalence of tachycardia in cirrhotic patients may impair effective QTc evaluation using the Bazett formula. Based on this fact, it is suggested to use different formulas in patients with cirrhosis, such as the Fridericia formula (QTc = QT/RR^1/3^) or a formula dedicated for cirrhotic patients (QTC = 453.65 × RR^1/3.02^). In recent research, the total number of identified QTc interval prolongation was 19% decreased by using Fridericia [45].

Electromechanical uncoupling in CCM patients leads to dyssynchrony between electrical and mechanical systole. There is a correlation between electromechanical uncoupling and QT interval prolongation, which increases the latency difference of electrical and mechanical contraction [46]. However, the clinical significance of these findings remains unclear [22].

Chronotropic incompetence is defined as an inability to proportionally increase HR as a response to increased metabolic demand, i.e., physical exertion or after dobutamine infusion. In early stages of cirrhosis the chronotropic incompetence may not be fully pronounced, probably due to desensitization and down-regulation of the ß-adrenergic receptor in cardiomyocyte [22].

### 2.4. Biomarkers

Natriuretic peptides—atrial (ANP), brain (BNP), and its prohormone (NT-proBNP)—are sensitive markers of myocardial injury, and their higher serum concentrations are associated with an advanced stage of cirrhosis, greater LV systolic dysfunction, myocardial hypertrophy and a poor prognosis [47].

ANP serum levels are elevated in patients with cirrhosis and ascites due to LA enlargement related to an increase in circulating blood volume [22].

BNP and NT-proBNP serum concentrations correspond to the liver failure stage, and it is highest in patients with class C in the Child–Pugh score and a high MELD score [39,48]. An elevated BNP serum concentration correlates with higher patients’ mortality within six months after discharge or after an OLTx procedure [38]. Abnormal NT-proBNP serum levels may be useful in the identification of patients with elevated cardiovascular risk. Further cardiological diagnostics are suggested in patients with an NT-proBNP concentration exceeding 290 pg/mL [22].

Troponins (Tn) are specific markers of myocardial necrosis, especially cTnT, if its area is limited. In some papers, it was found that higher cTnT serum concentrations in cirrhotic patients are correlated with an advanced stage of liver disease and higher mortality [49]. In our data, Tn I was not found to be a predicting marker of early cardiovascular morbidity in liver transplant recipients [50].

The TnI serum level is often elevated in patients with alcoholic liver cirrhotic and corresponds to impaired systolic function, although it is not related to an advanced stage of cirrhosis and portal hypertension [22].

Galectin-3 is a novel biomarker associated with cardiac diastolic [8] and systolic [51] dysfunction with documented clinical application in acute and chronic heart failure [52]. A significant increase in the galectin-3 serum level was found in patients with cirrhosis and in animal models of liver cirrhosis. However, this protein may also be secreted from other organs alongside the heart. In spite of this limitation, galectin-3 could be a potentially useful tool in the assessment of CCM [53].

The mechanisms of haemodynamic changes also include an imbalance between an overproduction of circulating vasodilators (adrenomedullin, tumor necrosis factor alpha (TNF-a)) and a reduction in levels of vasoconstrictors (endothelin-1 (ET-1)) [48].

### 2.5. Histopathology

Histologic changes of myocardial cells in cirrhotic cardiomyopathy are non-specific; they may be similar to those that are observed in alcoholic cardiomyopathy. They could look like areas of fibrosis, subendocardial edema. or vacuolation of the nucleus and cytoplasm [54].

### 2.6. Screening for CCM

As many studies have shown, liver failure significantly interferes with cardiovascular function. As a result, it may lead to hyperkinetic circulation, which results in cirrhotic cardiomyopathy. Therefore, it is extremely important to find an appropriate diagnostic algorithm for early diagnosis of CCM.

### 2.7. ECG

The first examination that should be performed by each patient is an electrocardiogram (ECG). In the group of patients suspected of hepatic cardiomyopathy, we are mainly looking for abnormalities in the form of a prolonged QT interval [45]. However, the presence of hyperkinetic circulation and the frequent occurrence of tachycardia, especially in the advanced stage of the disease, may hinder the actual evaluation of the QT interval corrected to heart rate. Therefore, in order to avoid inappropriate electrocardiographic evaluation (ECG), it is proposed that other forms, such as the Fridericia formula (QTc = QT/RR1/3) or the a formula for patients with cirrhosis (QTc = 453. 65 × RR1/3.02), be used to calculate the corrected QT interval in patients with cirrhosis. This is because, unlike the Bazett formula, they are less dependent on the length of the RR intervals [45].

### 2.8. Echocardiography

Another necessary examination is echocardiography. This is the basic method of detecting systolic and diastolic dysfunction of the heart muscle, which is essential for the diagnosis of CCM [40,55,56]. Patients with cirrhosis, even with CCM, have relatively rarely shown a decrease in the left ventricular ejection fraction (LVEF) in the resting test as a result of the presence of hyperkinetic circulation. In order to diagnose latent systolic dysfunction, it is proposed to perform a stress echocardiographic examination [57]. However, in the diagnosis of the diastolic dysfunction of the myocardium, doppler techniques are used [4,35]. Echocardiography with saline contrast helps diagnose the hepatopulmonary syndrome and portopulmonary hypertension in the case of patients with end stage liver disease (ESLD) with portal hypertension [58].

### 2.9. Biomarkers

In patients with cirrhosis we should determine the basic biomarkers, i.e., NT-proBNP and TnI [59,60,61]. Elevated levels of NT-proBNP and TnI are correlated with severity of liver disease. Due to the dilation of atria in the case of volume overload as a presentation of decompensation of liver disorder, the concentration of natriuretic peptides is dynamic and dependent on the effects of treatment of liver disorder.

### 2.10. Cardiopulmonary Exercise Test (CPET)

The Cardiopulmonary Exercise Test (CPET) is based on the analysis of respiratory gases during increasing exercise. It is a very good test in the assessment of physical capacity, as it enables the assessment of the maximum oxygen absorption at the peak of the exercise (VO2peak). Patients with liver failure are characterized by reduced physical capacity, and reduced oxygen absorption at the peak of the effort may result from many complications of the disease, including the presence of sarcopenia or CCM. A decrease in peak VO2 correlates with the severity of cirrhosis assessed by MELD and Child–Pugh scoring [62,63,64].

### 2.11. Magnetic Resonance Imaging (MRI)

Magnetic resonance imaging (MRI) allows for both morphological and functional evaluation of the heart muscle. It is a valuable addition to echocardiography, especially when it is limited by difficult technical conditions. MRI also assesses myocardial swelling and fibrosis in CCM. Observation of the entire cardiac cycle enables a precise functional analysis of the heart, including the assessment of left ventricular contractility disorders. It can therefore detect subclinical changes before manifest heart failure occurs. Unfortunately, the prevalence of diagnostics using this method is limited by its low availability and high cost [5,7,59,65].

### 2.12. CCM and Transjugular Intrahepatic Portosystemic Shunt

Substantial blood volume relocation from splanchnic vessels to the central circulation associated with the transjugular intrahepatic portosystemic shunt (TIPS) procedure may lead to rapid preload increase, aggravation of hyperdynamic circulation, and hemodynamic deterioration. There are several adjustments to the circulatory function after shunt establishment—increases in EDV, CO and pulmonary pressures. All of them normally resolve within 6–12 months after the procedure. Nevertheless, these alterations may become symptoms of progressive diastolic dysfunction in the course of CCM. Occurrence of symptomatic diastolic failure after a TIPS procedure is associated with worse ascites mobilization and poor prognosis [38]. However, the observed hemodynamic effects of TIPS depend on the circulatory system condition prior to procedure. There are singular case reports of acute heart failure and pulmonary oedema after a TIPS procedure [66].

### 2.13. CCM and Liver Transplantation

Orthotopic liver transplantation is a major stress factor for the circulatory system owing to significant alterations in both preload and afterload, as well as a massive release of mediators of inflammation and vasoactive agents [38,67,68]. Clamping of the inferior vena cava and portal vein, hemorrhage, fluids and blood infusion, as well as ischemia or reperfusion all cause hemodynamic alterations. CCM patients’ cardiovascular system is incapable to adapt to these changes, which may result in overt systolic and diastolic heart dysfunction. Liver recipients are at increased risk of developing cardiovascular complications, which occur in up to 70% of patients after OLTx [69]. The reperfusion phase of the allograft is burdened with the most cardiovascular instability [70]—postreperfusion syndrome (defined as a decrease in mean arterial pressure ≥30% below the baseline value, lasting for at least 1 min, occurring during the first 5 min after reperfusion of the graft), which occurs in 12–77% of recipients [71]. Moreover, approximately up to 7% of deaths after OLTx are associated with cardiac causes [72].

The perioperative deviations in preload, afterload, and peripheral vascular resistance may reveal impaired myocardial contractile responsiveness [32]. More than 70% of liver transplant recipients may experience one or more cardiovascular-related complications after liver transplantation [73]. Different studies have shown that in 23–35% of patients after graft reperfusion there occurs a decrease in systolic function expressed by reduced CO [22,74]. Pulmonary oedema confirmed by clinical manifestations or imaging has been reported in 12% to 56% of patients during the first postoperative week after liver transplantation [75]. Nevertheless, the cardiac morphology and systolic function reverted to normal within 6 to 12 months after OLTx [11,76].

The role of diastolic dysfunction in predicting or determining the prognosis of patients with cirrhosis remains unclear. Three independent studies revealed significant diastolic dysfunction in patients after OLTx [12,35,77], although another study determined hemodynamic improvement as a result of the transplantation [75]. The impact of the diastolic disfunction on the patient survival after OLTx is also controversial. One study found that the presence of diastolic disfunction was not correlated with worse long-term adverse outcomes after transplantation [35,67], although other studies have claimed that patients with diastolic dysfunction have significantly lower survival rates compared to those without diastolic dysfunction over a two-year follow-up period [78,79]. Furthermore, patients with higher grades of diastolic dysfunction are at increased risk of perioperative heart failure [32]. It has also been suggested that pre-transplant diastolic dysfunction is associated with an increased risk of allograft rejection, grant failure, and left ventricular hypertrophy [80,81]. These findings correlated with time-dependent irreversible fibrosis might be the reason for the increased LV mass index observed in patients after OLTx. However, this increase in ventricular mass remains controversial and unclear [82]. Another study underlined the possible role of a high level of BNP as a predictor of diastolic dysfunction and increased mortality after OLTx [60]. The diastolic dysfunction, similar to systolic dysfunction, tends to return to normal between 6 and 12 months after OLTx [75,76]. All of these data indicate a demand for a careful pre-transplant cardiac assessment of cirrhotic patients in the direction of CCM [83].

Liver transplantation also impacts electrical heart activity. Amelioration of electrophysiological dysfunction displayed by a decline of the QTc interval prolongation occurred in 55–72% of the patient population after OLTx [38,42].

Unfortunately, to date no guidelines for CCM management have been developed. Despite their limitations, studies consistently indicate an improvement in cardiovascular performance after liver transplantation, therefore establishing that transplantation may be a causal treatment of CCM [32,68]. However, in the era of organ shortage, cirrhotic cardiomyopathy is unlikely to be the sole indication for liver transplantation. It appears that accurate pre-, peri-, and post-transplant management is crucial in the improvement of patients’ prognosis. In order to meet these demands, VanWagner and colleagues have presented a point-based prediction model for cardiovascular risk in OLTx—The CAR-OLT score, which could be a useful addition for the prediction of cardiovascular events and outcomes after transplantation [84].

## 3. Therapy

As there is no specific treatment for this disease, the therapy should focus on symptomatic treatment of congestive heart failure. Diuretics are mainly recommended. It is also recommended to use beta-blockers to reduce the risk of bleeding from esophageal varices, as it is a frequent complication of cirrhosis and portal hypertension. In addition, administration of a single dose of beta-blockers improves the QT interval in cirrhosis. However, chronic administration of non-selective beta-blockers is effective in shortening the QT interval only in patients with an initially prolonged QT interval [85].

Drugs from the group of angiotensin-converting enzyme inhibitors are contraindicated due to a further decrease in peripheral resistance, which adversely affects the haemodynamic state in patients with CCM. However, angiotensin II receptor blockers (ARBs) may show beneficial effects due to reducing the left ventricular dimension, reducing myocardial hypertrophy, and improving diastolic function [4].

## 4. Gaps of Knowledge

The most important issue is to update the definition of hepatic cardiomyopathy and the criteria for diagnosis. In addition, the diagnostic methods in the CCM should be evaluated and harmonized. Better understanding of the pathogenesis and pathology of cirrhotic cardiomyopathy is crucial in developing more accurate diagnostic tools and specific treatments for this condition. Another important point for further research is the prevalence of CCM—the impact of the aetiology of liver cirrhosis is still unknown. There are also limited data on the association between CCM and hepatopulmonary syndrome as well as other specific liver cirrhosis complications.

We should also consider methods for the early diagnosis of cardiovascular dysfunction in patients with cirrhosis. Earlier diagnosis may have prognostic significance for patients, and it may be associated with lower risk of irreversible cardiovascular dysfunction and complications and with longer survival.

Rules for treatment in patients with CCM should be developed. Liver transplantation is currently the only effective method of treatment of liver cardiomyopathy and prolonged survival. Since the operation itself involves a very high cardiovascular risk, a detailed cardiological evaluation of patients and their proper qualification and preparation for liver transplantation should be developed. After liver transplantation, we should also include patients in special post-operative care due to their general condition and increased risk of complications. It is also important to provide an individual rehabilitation plan after the operation, in cooperation with physiotherapists. There is a demand to assess novel biomarkers indicating specific changes in cirrhotic patients’ circulation and include their evaluation in therapeutic routines and risk stratification.

## 5. Conclusions

Since liver cardiomyopathy is usually asymptomatic, and any symptoms are very unspecific and may overlap with the symptoms of the underlying disease, we should observe and evaluate patients particularly carefully for cardiovascular abnormalities. Clinical understanding of CCM in end-stage liver disease is the best way to avoid life-threatening consequences in patients undergoing special procedures (TIPS or OLT).

The pathophysiology and importance of CCM have been of interest to many scientists over the past decades. However, not all issues concerning CCM have been clarified to this day, so further research is needed on this condition.

Despite newly proposed diagnostic criteria there still are many additional criteria that are not fully understood. Moreover, there is a demand to find new diagnostic tools for further CCM investigations, such as novel biomarkers or new cardiac imaging methods.

Additionally, the new CAR-OLT score should be evaluated in a higher number of patients undergoing OLTx. A multidisciplinary approach to CCM is crucial for understating its clinical impact on patients with ESLD and undergoing transplantation.

Taking into account the abovementioned problems, preparing new scientific papers remains very necessary and urgent for the verification and organization of knowledge concerning CCM.

## Figures and Tables

**Figure 1 jpm-11-01285-f001:**
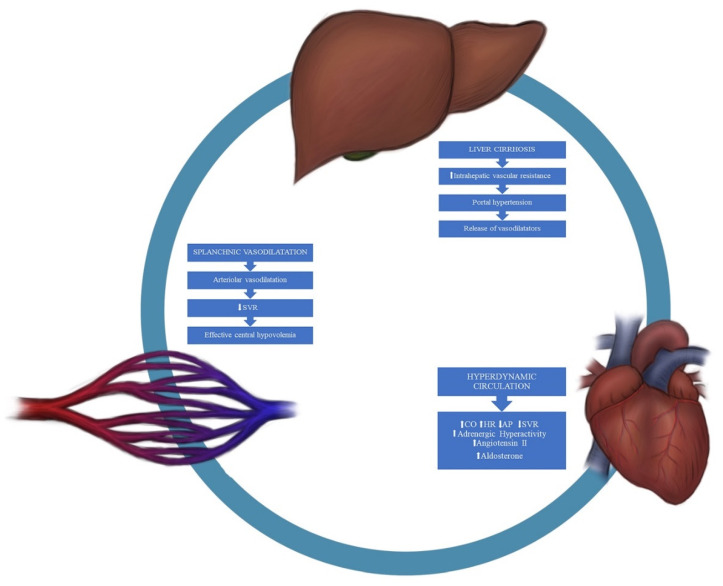
Patomechanism of CCM. SVR—systemic vascular resistance; CO—cardiac output; HR—heart rate; AP—arterial pressure.

**Table 1 jpm-11-01285-t001:** Comparison of 2005 and 2019 CCM criteria [7]. GLS—global longitudinal strain; LV—left ventricle; E—early diastolic transmitral filling; A—late diastolic transmitral filling; e’—early diastolic mitral annular velocity; LAVI—left atrium volume index; TR—tricuspid regurgitation; QTc—corrected QT interval; BNP—brain natriuretic peptide; CO—cardiac output; SVR—systemic vascular resistance; HR—heart rate; SV—stroke volume; 6MWT—6-min walk test; CPET—cardio-pulmonary exercise test; MRI—magnetic resonance imaging.

CCM Diagnostic Criteria	
I. Systolic Dysfunction	
2005 criteria: (any of the following)	2019 proposed criteria: (any of the following)
LV ejection fraction < 55% Blunted contractile response on stress testing	LV ejection fraction ≤ 50% Absolute ^1^ GLS < 18%
AND/OR II. Diastolic dysfunction	
2005 criteria: (any of the following)	2019 proposed criteria: (≥3 of the following)
Deceleration time > 200 ms Isovolumetric relaxation time > 80 ms E/A < 1	Septal e′ velocity < 7 cm/s E/e′ ratio ≥ 15 LAVI > 34 mL/m^2^ TR velocity ^2^ > 2.8 m/second
IIIa. Supportive criteria (Not diagnostic!)	IIIb. Areas for further research (Validation required)
2005 criteria:	2019 proposed criteria:
Electrophysiological abnormalities Abnormal chronotropic response Electromechanical uncoupling Prolonged QTc interval Enlarged left atrium Increased myocardial mass Increased BNP Increased proBNP Increased troponin I	Abnormal chronotropic or inotropic response ^3^ Electrocardiographic changes Electromechanical uncoupling Myocardial mass change Serum biomarkers Chamber enlargement Cardiac MRI ^4^ Our proposal: Haemodynamic parameters (CO, SVR, HR, SV) Exercise test (6MWT or CPET)

## Data Availability

Data available in a publicly accessible repository.

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
