# Peer review of "What’s New in Cirrhotic Cardiomyopathy?—Review Article"

_jpm, 2021, doi:10.3390/jpm11121285_

Round 1

Reviewer 1 Report

This is an interesting article on a controversial topic.

As a review article, there should be a greater development of the pathophysiological chapter, which is presented here in a very brief way.

In the chapter CCM and liver transplantantion - despite the majority of cardiac dysfunctions after liver transplantation have to do with diastolic dysfunction, there is also an increase in the LV mass index that should be specified and explained why.

Author Response

Dear Reviewers,

Thank you for the discerning comments concerning our manuscript entitled “What’ new in cirrhotic cardiomyopathy – review article” which we accept with comprehension and gratitude. We have studied your comments carefully and made corrections which we hope will meet with your approval. Your questions or comments are answered in detail below, with original reviewer comments denoted in boldface, our responses in regular typeface and all changes in the manuscript in red font.

We would like to thank you for kind words and we corrected the manuscript according to your advices.

Responses to reviewers:

Reviewer 1

  1. As a review article, there should be a greater development of the pathophysiological chapter, which is presented here in a very brief way.

Response: Thank you for your advice. The pathosiological chapter is is divided into sections where the pathophysiology of hepatic cardiomyopathy is explained in more detail. We have emphasized this in the text to make it more readable. We also added short paragraph about histopathology.

Change: Pathomechanisms underlying cirrhotic cardiomyopathy are complex and not clearly understood. Moreover, their prevalence in course of disease is also hard to determine. Cirrhotic cardiomyopathy is characterized by an impaired cardiac response to stress, which results from a combination of autonomic dysfunction, alterations in cell membrane composition, ion channel defects, and an overproduction of cardio depressant factors.  A more detailed pathomechanism is presented in the following sections, and the diagram is presented in Figure 1.

Histopathology

Histologic changes of myocardial cell in cirrhotic cardiomyopathy are non-specific, they may be similar to those which are observed in alcoholic cardiomyopathy. They could look as areas of fibrosis, subendocardial edema or vacuolation of the nucleus and cytoplasm.

  1. In the chapter CCM and liver transplantantion - despite the majority of cardiac dysfunctions after liver transplantation have to do with diastolic dysfunction, there is also an increase in the LV mass index that should be specified and explained why.

Response: This finding is also associated with pre-transplant diastolic dysfunction which is associated with myocardial fibrosis, increased ventricular stiffness and subendothelial edema. This time dependent increase in fibrosis which cannot be reversed may explain the observation of increased LV mass index.

Change: We added an insight on this topic in chapter about liver transplantation

We also asked for opinion of native English speaker and applied some alterations in text.

Reviewer 2 Report

1. About the Abstract:

I suggest to re-do the Abstract following the “Instruction for the Authors” like this form:

Abstract should follow the style of structured form, but without headings: 1) Background: Place the question addressed in a broad context and highlight the purpose of the study; 2) Methods: Describe briefly the main methods or treatments applied. Include any relevant preregistration numbers, and species and strains of any animals used. 3) Results: Summarize the article's main findings; 4) Conclusion: Indicate the main conclusions or interpretations.

2.About the Background. It could be better explained about 2.019 Proposed Criteria.

 The comments and references must be included below the tables.

3.About the Methods: I suggest that it could be written in ¨Methods¨ how the authors found out the best studies to compose their review study, citing what basis of dates were used and how the authors have done a chronologic comparison of the definitions of cirrhotic cardiopathy through the timeline with the intension to improve that definition.

4.Number 1 (LINE 32). The number 1 must be excluded: that is no other numerical number in this current text

5.MOBILISATION (line 301): the  corret for is ¨MOBILIZATION¨

6.ANOTHER (LINE (334): the correct form is ¨OTHER¨

7.SARTANS (line 368): I did not understand what does it mean

8. About the ¨References¨: As said in “Instructions for the Authors” “References must be numbered in order of appearance in the text (including table captions and figure legends) and listed individually at the end of the manuscript”. So in “Epidemiology and diagnostic criteria” (line 67), the numbers of the References are wrong. The same with the References [79], [80], [81], [84], etc.

9. About the ¨Conclusions¨: I suggest that this part may include ¨what is new¨ about the present paper theme, answering the title´s question of your paper. It should emphasize the new diagnosis criteria.

Author Response

Dear Reviewers,

Thank you for the discerning comments concerning our manuscript entitled “What’ new in cirrhotic cardiomyopathy – review article” which we accept with comprehension and gratitude. We have studied your comments carefully and made corrections which we hope will meet with your approval. Your questions or comments are answered in detail below, with original reviewer comments denoted in boldface, our responses in regular typeface and all changes in the manuscript in red font.

We would like to thank you for kind words and we corrected the manuscript according to your advices

Reviewer 2

  1. About the Abstract:

I suggest to re-do the Abstract following the “Instruction for the Authors” like this form:

Abstract should follow the style of structured form, but without headings: 1) Background: Place the question addressed in a broad context and highlight the purpose of the study; 2) Methods: Describe briefly the main methods or treatments applied. Include any relevant preregistration numbers, and species and strains of any animals used. 3) Results: Summarize the article's main findings; 4) Conclusion: Indicate the main conclusions or interpretations.

Response: Thank you for this advice. However, our article is a review article which hardly fits the structure typical to the original papers.

Change: We adjusted the abstract form accordingly to the proposed structure.

2.About the Background. It could be better explained about 2.019 Proposed Criteria.

 The comments and references must be included below the tables.

Response: Thank you for the advice. We agree.

Change: We added a description of table 1. The new criteria were described in the “Epidemiology and diagnostic criteria” part. We added more insight to it.

3.About the Methods: I suggest that it could be written in ¨Methods¨ how the authors found out the best studies to compose their review study, citing what basis of dates were used and how the authors have done a chronologic comparison of the definitions of cirrhotic cardiopathy through the timeline with the intension to improve that definition.

Response: Thank you for your advice. We added “methods” accordingly to your suggestion

Change: Methods

We performed a review of the literature. The search was carried using PubMed and ScienceDirect as data sources. We searched classical articles and clinical trials. The following keywords were used: “cirrhotic cardiomyopathy”. The inclusion criteria included articles which reported cirrhotic cardiomyopathy written in the years 1953-2021. Non-English articles were excluded.

4.Number 1 (LINE 32). The number 1 must be excluded: that is no other numerical number in this current text

Response: We agree

Change: We changed accordingly to your suggestion

5.MOBILISATION (line 301): the  corret for is ¨MOBILIZATION¨

Reponse: Thank you for your advice

Change: We changed the misspelling

6.ANOTHER (LINE (334): the correct form is ¨OTHER¨

Reponse: Thank you for your advice

Change: We changed the misspelling

7.SARTANS (line 368): I did not understand what does it mean

Response: Thank you for your advice

Change: We changed it: angiotensin II receptor blockers (ARBs)

  1. About the ¨References¨: As said in “Instructions for the Authors” “References must be numbered in order of appearance in the text (including table captions and figure legends) and listed individually at the end of the manuscript”. So in “Epidemiology and diagnostic criteria” (line 67), the numbers of the References are wrong. The same with the References [79], [80], [81], [84], etc.

Response: We agree

Change: We changed accordingly to your suggestion

  1. About the ¨Conclusions¨: I suggest that this part may include ¨what is new¨ about the present paper theme, answering the title´s question of your paper. It should emphasize the new diagnosis criteria.

Response: We agree.

Change: We added more insight on new findings and new perspectives in further research.

We also asked for opinion of native English speaker and applied some alterations in text.